# Star Temporal Classification: Sequence Modeling with Partially Labeled Data

**Vineel Pratap**
Meta AI

**Awni Hannun**[*]
Zoom AI

**Gabriel Synnaeve**
Meta AI

**Ronan Collobert**[*]
Meta AI

## Abstract

We develop an algorithm which can learn from partially labeled and unsegmented sequential data. Most sequential loss functions, such as Connectionist Temporal Classification (CTC), break down when many labels are missing. We address this problem with Star Temporal Classification (STC) which uses a special *star* token to allow alignments which include all possible tokens whenever a token could be missing. We express STC as the composition of weighted finite-state transducers (WFSTs) and use GTN (a framework for automatic differentiation with WFSTs) to compute gradients. We perform extensive experiments on automatic speech recognition. These experiments show that STC can close the performance gap with supervised baseline to about 1% WER when up to 70% of the labels are missing. We also perform experiments in handwriting recognition to show that our method easily applies to other sequence classification tasks.

## 1 Introduction

Applications of machine learning in settings with little or no labeled data are growing rapidly. Much prior work in temporal classification focuses on semi-supervised or self-supervised learning [6, 37]. Instead, we focus on weakly supervised learning, in which the learner receives incomplete labels for the examples in the data set. Examples include multi-instance learning [1, 44], and partial-label learning [18, 9, 25]. In our case, we assume that each example in the training set is partially labeled, as in Figure 1: neither the number of missing words nor their positions in the label sequence are known in advance.

Input: 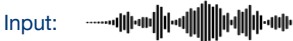

Original Label: seen from an airplane the island looks like a big spider

Partial Label: from airplane the a spider

Figure 1: An example of speech with a complete and a partial label.

Partial labeling problem could arise in many practical scenarios. In semi-supervised learning or transfer learning of sequence labeling tasks, we can only consider high confidence tokens in the label sequence and discard rest of the tokens. These labels with incomplete tokens can be treated as partial labels. When transcribing a dataset, we can randomly label parts of the dataset and use them as partial labels, at a much lower cost than transcribing the full dataset. We can also consider semi-automatic ways to label a dataset. Restoring damaged texts [2, 3] where some of the writing is illegible could be considered as a partial labeling problem. We also often face partial alignment problem [5] in practice

---

[*]Currently at Apple. Correspondence to Vineel Pratap ⟨vineelkpratap@meta.com⟩.

36th Conference on Neural Information Processing Systems (NeurIPS 2022).

where beginning and ending tokens are missing. This is a sub-problem of the more general version of partial labeling problem that we solve in this work.

To solve this weak supervision problem we develop the Star Temporal Classification (STC) loss function. The STC algorithm explicitly accounts for the possibility of missing labels in the output sequence at any position, through a special *star* token. For the example of Figure 1, the STC label sequence is `* from * airplane * the * a * spider *` (where `*` can be anything but the following label token). We show how to simply implement STC in the weighted finite-state transducer (WFST) framework. Moreover, through the use of automatic differentiation with WFSTs, gradients of the loss function with respect to the inputs are computed automatically, allowing STC to be used with standard neural network training pipelines.

For practical applications, a naive implementation of STC using WFSTs is intractably slow. We propose several optimizations to improve the efficiency of the STC loss. The *star* token we use results in much smaller graphs since it combines many arcs into one. The *star* token also enables us to substantially reduce the amount of memory transfers between the GPU, on which the network is computed, and the CPU, on which the STC algorithm is evaluated. These optimizations result in only a minor increase in model training time when using the STC loss over a CTC baseline.

The STC loss can be applied to most sequence transduction tasks including speech recognition, handwriting recognition, machine translation, and action recognition from videos. Here we demonstrate the effectiveness of STC on both automatic speech recognition and offline handwriting recognition. In speech recognition, with up to 70% of the labels missing, STC is able to achieve similar (about 1% WER difference) performance as supervised baseline . Similarly in handwriting recognition STC can achieve low character error rates with up to 50% of the labels missing. In both cases STC yields dramatic gains over a baseline system which does not explicitly handle missing labels.

To summarize, the main contributions of this work are:

- We introduce STC, a novel temporal classification algorithm which explicitly handles an arbitrary number of missing labels with an indeterminate location.
- We simplify the implementation of STC by using WFSTs and automatic differentiation.
- We describe key optimizations for STC enabling it to be used with minimal increase in training time.
- We show in practical speech and handwriting recognition tasks that STC yields low word and character error rates with up to 70% of the labels missing.

## 2 Related Work

Many generalizations and variations of CTC have been proposed. Graph-based temporal classification (GTC) [32] generalizes CTC to allow for an $n$-best list of possible labels. [40] also extends CTC for using with for multiple labels. [23] propose selfless-CTC which disallows self-loops for non-blank output tokens. Unlike these alternatives, STC allows for an arbitrary number of missing tokens anywhere in the label for a given example. A recent work, concurrent with this work, proposes a wild card version of CTC [5]. However, in that work the wild cards, or missing tokens, can only be at the beginning or end of the label.

Extended Connectionist Temporal Classification (ECTC) [17] was developed for weakly supervised video action labeling. However, the weak supervision in this case refers to the lack of a segmentation, which is the common paradigm for CTC. The extensions to CTC are intended to handle the much higher ratio of input video frames to output action labels. [11] develop a sequence level loss function which can learn from noisy labels. They extend the auto segmentation criterion [7] to include a noise model which explicitly handles up to one missing label between tokens in the output. Unlike that work, STC extends the more commonly used CTC loss and allows for any number of missing tokens.

The idea of expressing CTC as the composition of WFSTs is well known [27, 23, 41]. However, recently developed WFST-based frameworks which support automatic differentiation [15, 19] make the development of variations of the CTC loss much simpler.

WFSTs have been modified by to take into account of noisy transcripts [16, 31, 4] by composing the WFST graphs of word sequences. However, in this work the WFST composition is between

significantly larger WFST graphs (label sequence and neural network model output sequence). Also, the composition is performed during the training of the neural network which makes it a more challenging engineering problem.

## 3  Method

### 3.1  Problem Description

We consider the problem of temporal classification [20], which predicts an output sequence $\mathbf{y} = [y_1, ...., y_U] \in \mathcal{A}^{1 \times U}$, where $\mathcal{A}$ is a fixed alphabet of possible output tokens, from an unsegmented input sequence $\mathbf{x} = [x_1, ..., x_T] \in \mathbb{R}^{d \times T}$, where each $x_i$ is a $d$-dimensional feature vector. We also assume that the length of the output $U$ is less than or equal to that of the input, $T$. A partially labeled output sequence $\tilde{\mathbf{y}}$ of length $\tilde{U}$, such that $(\tilde{U} \leq U)$, is formed by removing zero or more elements from the true output sequence $\mathbf{y}$. Given a training set $\mathcal{D} \triangleq \{\mathbf{x}^i, \tilde{\mathbf{y}}^i : i = 1, \ldots, N\}$ consisting of input sequences $\mathbf{x}^i = [x_1, \ldots, x_{T^i}]$ and partially labeled output sequences $\tilde{\mathbf{y}}^i = [y_1, \ldots, y_{\tilde{U}^i}]$ our goal is to learn temporal models which can predict true output sequences $\mathbf{y}^i$. We consider this a weakly supervised temporal classification problem as the learning algorithm only has access to a subset of the true output labels during training.

### 3.2  Connectionist Temporal Classification (CTC)

Connectionist Temporal Classification (CTC) [13] is a widely used loss function for training sequence models where the alignment of input sequence with the target labels is not known in advance.

Consider an input sequence $\mathbf{x} = [x_1, \ldots, x_T]$ and target sequence $\mathbf{y} = [y_1, \ldots, y_U] \in \mathcal{A}^{1 \times U}$, where $\mathcal{A}$ is a finite alphabet of possible output tokens and $U \leq T$. The CTC loss uses a special *blank* token, $\langle b \rangle$, to represents frames which do not correspond to an output token. An alignment between the input $\mathbf{x}$ and output $\mathbf{y}$ in CTC is represented by a sequence of length $T$; $\boldsymbol{\pi} = [\pi_1, \pi_2, ..., \pi_T]$ where $\pi_t \in \mathcal{A} \cup \{blank\}$. The CTC collapse function $\mathcal{B}$ maps an alignment to an output, $\mathcal{B}(\pi) = \mathbf{y}$. The function $\mathcal{B}$ removes all but one of any consecutively repeated tokens and then removes *blank* tokens e.g., (a, b, *blank*, *blank*, b, b, *blank*, a) $\mapsto$ (a, b, b, a). Given the input $\mathbf{x}$, each output is independent of all other outputs, hence the probability of an alignment $P(\boldsymbol{\pi}|\mathbf{x})$ is:

$$P(\boldsymbol{\pi}|\mathbf{x}) = \prod_{i=1}^{T} P(\pi_t|\mathbf{x}). \tag{1}$$

There can be many possible alignments $\boldsymbol{\pi}$ for a given $\mathbf{x}$ and $\mathbf{y}$ pair. The CTC loss computes the negative log probability of $\mathbf{y}$ given $\mathbf{x}$ by marginalizing over all possible alignments:

$$L_{CTC} = -\log P(\mathbf{y}|\mathbf{x}) = -\log \sum_{\boldsymbol{\pi} \in \mathcal{B}^{-1}(\mathbf{y})} P(\boldsymbol{\pi}|\mathbf{x}). \tag{2}$$

The sum over all alignments can be computed efficiently with dynamic programming as described by [13].

The CTC loss can be computed purely with WFST[2] operations as described in Figure 2. The composition of the emission graph (Figure 2a) which has arc weights weights corresponding to $\log P(\pi_t|\mathbf{x})$ and the label graph (Figure 2b) is shown in Figure 2c. Each path from a start to a final state in Figure 2c is a valid CTC alignment. The negation of the forward score of the composed graph gives the CTC loss. These operations are differentiable and gradients can easily be propagated backward from the loss to a neural network to train the model.

Constructing the CTC loss from simpler WFST graphs also simplifies the implementation of variations of CTC. For example, the variations of [32] only require changes to the label graph $\mathcal{Y}_{ctc}$. However, when efficiency is a primary concern, custom GPU kernels can be faster than a generic implementation with WFSTs.

---

[2]A brief introduction to WFSTs can be found in the Appendix A

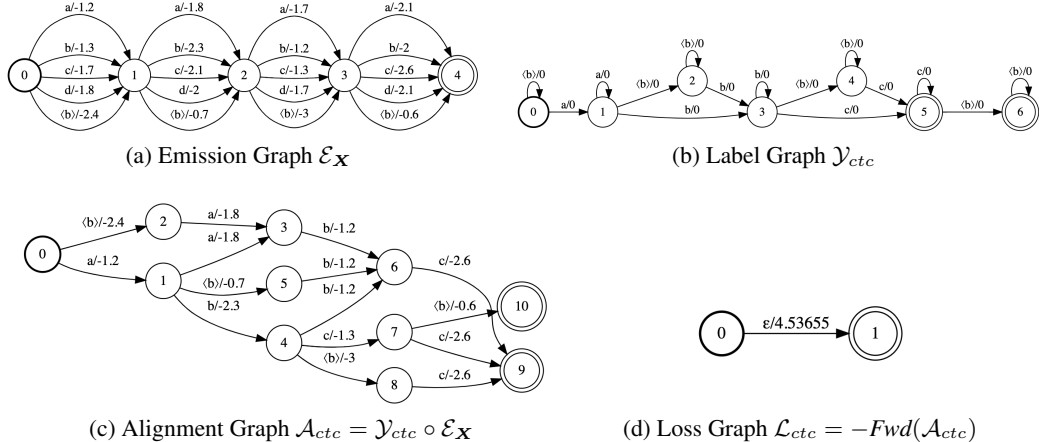

(a) Emission Graph $\mathcal{E}_{\boldsymbol{X}}$        (b) Label Graph $\mathcal{Y}_{ctc}$

(c) Alignment Graph $\mathcal{A}_{ctc} = \mathcal{Y}_{ctc} \circ \mathcal{E}_{\boldsymbol{X}}$      (d) Loss Graph $\mathcal{L}_{ctc} = -Fwd(\mathcal{A}_{ctc})$

Figure 2: The sequence of steps involved in computing CTC loss using WFSTs. The arc label "p/w" is a shorthand notation for "p:p/w". (a) is the emissions graph constructed from the log probabilities over the alphabet $\mathcal{A} = \{a, b, c, d\}$ and blank symbol $\langle b \rangle$. The CTC label graph corresponding to the target sequence $(a, b, c)$ is shown in (b). In (c), we compose label graph with emission graph to get all valid paths which collapse to the target sequence. Finally, in (d) we sum the probabilities of all of the valid paths (in log-space) and negate the result to yield the CTC loss.

### 3.3 Star Temporal Classification (STC)

The label graph of CTC, $\mathcal{L}_{ctc}$ (Figure 2b), constructed from partial labels does not allow for the true target as a possibility. The STC algorithm addresses this problem by allowing for zero or more tokens from the alphabet between any two tokens in the partial label. Like CTC, an alignment between the input $\mathbf{x}$ and output $\mathbf{y}$ in STC is represented by a sequence of length $T$; $\boldsymbol{\pi} = [\pi_1, \pi_2, ..., \pi_T]$ where $\pi_t \in \mathcal{A} \cup \{blank\}$ and $\mathbf{y}$ is a partial label of $\mathcal{B}'(\pi)$. Unlike CTC, the STC collapse function $\mathcal{B}'(\pi)$ only removes blank tokens. In other words, we do not allow self-loops on non $blank$ tokens in the STC label graph. This enables us to use a token insertion penalty as discussed in Section 3.3.1.

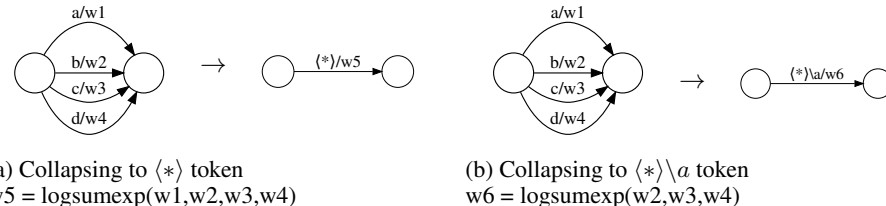

(a) Collapsing to $\langle * \rangle$ token
w5 = logsumexp(w1,w2,w3,w4)

(b) Collapsing to $\langle * \rangle \backslash a$ token
w6 = logsumexp(w2,w3,w4)

Figure 3: Example showing the collapsing of tokens in alphabet $\mathcal{A} = \{a, b, c, d\}$ to star token, $\langle * \rangle$

To make the computation of the STC loss efficient, STC uses a special *star* token, $\langle * \rangle$, which represents every token in the alphabet. It can be used to collapse the arcs between any two nodes in the WFST graph as shown in Figure 3a. We also define $\langle * \rangle \backslash t = \{y : y \in \mathcal{A}; y \neq t\}$ which is the relative complement of $t$ in $\mathcal{A}$. We use the token, $\langle * \rangle \backslash t$, in the label graph of STC, $\mathcal{L}_{stc}$, to avoid counting the same alignment multiple times as dicsussed in Section 3.3.2 The STC label graph, $\mathcal{L}_{stc}$, for the partial label $(a, b, c)$ is shown in Figure 4. The graph allows any alignment with zero or more tokens in between any two tokens of the given partial label sequence.

We also manually add the $\langle * \rangle$ and $\langle * \rangle \backslash t$ tokens to the emission graph $\mathcal{E}_{\boldsymbol{X}}$ with weights corresponding to their log-probabilities for each timestep $\mathbf{t}$ given by:

$$P(\langle * \rangle | \mathbf{x}, \mathbf{t}) = \sum_{y \in \mathcal{A}} P(y | \mathbf{x}, \mathbf{t}) = 1 - P(\langle b \rangle | \mathbf{x}, \mathbf{t})$$

$$P(\langle * \rangle \backslash t | \mathbf{x}, \mathbf{t}) = \sum_{y \in \mathcal{A}; y \neq t} P(y | \mathbf{x}, \mathbf{t}). \tag{3}$$

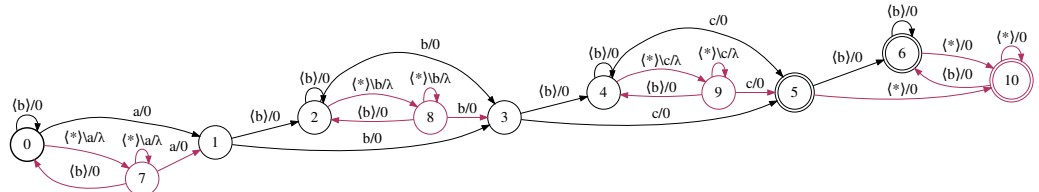

Figure 4: STC Label Graph, $\mathcal{Y}_{stc}$ for the output sequence $(a, b, c)$. In regex terms, this corresponds to "$[^a] * a\, [^b] * b\, [^c] * c\, .*$". $\langle * \rangle$, $\langle b \rangle$ refer to *star*, *blank* tokens and $\langle * \rangle \backslash a$ is the relative complement of $a$ in $\langle * \rangle$. $\lambda (\leq 0)$ corresponds to token insertion penalty which has a regularization effect. The arc transitions marked in red show the changes from a selfless-CTC label graph.

### 3.3.1 Token Insertion Penalty, $\lambda$

We noticed that model training with STC does not work if we train directly with the STC label graph, even when $10\%$ of the labels are missing. This is because the new paths introduced in STC from CTC (marked in red in Figure 4) would allow the model to produce a lot of possible output sequences for any given example and can confuse the model, especially during the early stages of training. To circumvent this problem, we use a parameter called token insertion penalty, $\lambda$ to add a penalty when using lot of new tokens. We define STC loss in the following way

$$\text{STC Loss} = -\log(p^{k_1} P_{path_1} + p^{k_2} P_{path_2} + ... + p^{k_N} P_{path_N})$$

where $P_{path_i}$ is the probability of an alignment path $i$, $p \in (0, 1]$ and $k_i$ is the number of additional star tokens for path $i$. This down weight paths which include many additional tokens. The token insertion penalty is in log-space and is defined as $\lambda = \log(p)$.

We use a exponential decay scheme to gradually reduce the penalty as the model starts training.

$$\begin{aligned} p_t &= p_{max} + (p_0 - p_{max})exp(-t/\tau) \\ \lambda_t &= ln(p_t) \end{aligned} \tag{4}$$

where $p_0, p_{max}, \tau$ are hyperparameters and $\lambda_t$ denotes the token insertion penalty used in STC for training step $t$. For choosing value of $\tau$, it is useful think in terms of half-life $t_{1/2} = \tau ln(2)$, which is the number of time steps taken for $p_t$ to reach $(p_0 + p_{max})/2$

### 3.3.2 On the importance of $\langle * \rangle \backslash t$ token

The "$\langle * \rangle \backslash t$" token which matches any token except the token, $t$ is useful to not count the same alignment path multiple times while computing the STC loss. We illustrate it through the two possible STC label graphs from Figure 5. Consider that the partial label has just one word "the" and number of time steps, T = 3. In Figure 5a, the alignment path {"the", "the", "cat"} is counted multiple times when we use start token $\langle * \rangle$ directly $0 \rightarrow 1 \rightarrow 2 \rightarrow 4$ and $0 \rightarrow 2 \rightarrow 4 \rightarrow 4$. However, if we use $\langle * \rangle \backslash the$ token as shown in Figure 5b , we do not have this issue of accepting the path "the the cat" multiple times.

It should also be noted that the "$\langle * \rangle \backslash t$" token is only present after the token $t$ but not before the token $t$ in the label graph of STC from Figure 4. So, the label graph would still accept all possible labels sequences for the given partial label.

### 3.3.3 Implementation Details

With the recent advent of frameworks such as GTN[15], K2[19] which support autograd on WF-STs, we focus on the engineering details used by us to use these frameworks and train large-scale recognition systems below.

We use the CPU-based WFST algorithms from GTN [15] to implement the STC criterion. Multiple threads are used to compute STC loss in parallel for all of the examples in a batch. Since the neural network model is run on GPU, the emissions needed by STC must be copied to the CPU, and the STC gradients must be copied back to GPU. To reduce the amount of data transfer between the CPU and the GPU, we transfer values corresponding only to the tokens present in the partial labels and the *star* tokens. This works since the gradients corresponding to all other tokens are zero.

Figure 6 shows an overview of the STC training pipeline. The input sequence is typically passed through a neural network on GPU to produce a frame wise distribution (in log-space) over $\mathcal{A} \cup \{blank\}$. The output number of frames depend on the size of the input sequence, as well as the amount of padding and the stride of the neural network model architecture. We then compute the log-probability of star token, $\langle * \rangle$ and $\langle * \rangle \backslash t$ tokens $\forall t \in \mathbf{y}$ using Equation 3. We construct the emission graph $\mathcal{E}_{\mathbf{X}}$ with weights as log-probabilities and the label graph $\mathcal{L}_{stc}$ on CPU. Using the WFST operations shown in Figure 2, we compute STC loss and gradients of loss with respect to arc weights (log probability output) of $\mathcal{E}_{\mathbf{X}}$. These gradients are copied back to GPU and the gradients of parameters in neural network are computed using backpropagation. Then the model can be trained using standard gradient descent methods.

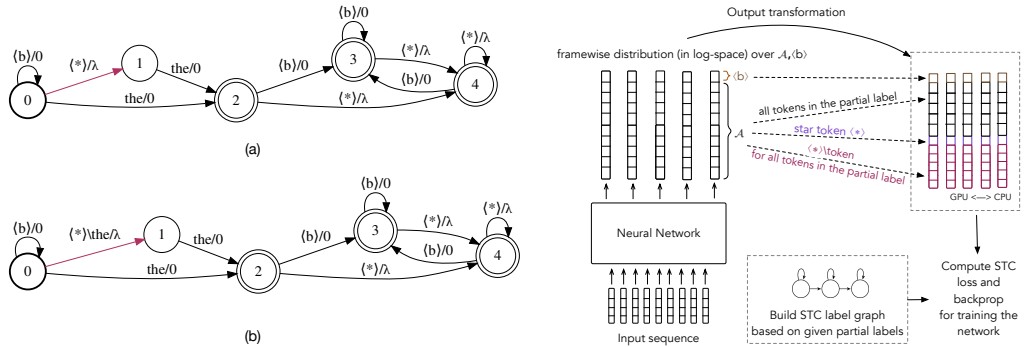

Figure 5: Examples to illustrate the importance of $\langle * \rangle \backslash t$ token

Figure 6: The STC Training Pipeline

# 4 Experimental Setup

## 4.1 Automatic Speech Recognition (ASR)

We use LibriSpeech [33] dataset, containing 960 hours of training audio with paired transcriptions for our speech recognition experiments. The standard LibriSpeech validation sets (*dev-clean* and *dev-other*) are used to tune all hyperparameters, as well as to select the best models. We report final word error rate (WER) performance on the test sets (*test-clean* and *test-other*).

We use the top 50K words (sorted by occurance frequency) from the official language model (LM) training data provided with LibriSpeech as the alphabet for training the models. These 50K words cover 99.04% of all the word occurances in the LM training data. The model architectures and the STC loss are implemented with the ASR application [35] of the flashlight[3] machine-learning framework and the C++ API of GTN.

### 4.1.1 Label Generation for Weakly Supervised Setup

As LibriSpeech dataset consists of fully labeled data, we need to drop the labels manually to simulate the weakly supervised setup. We quantify this using a parameter $p\_drop \in [0, 1]$, which denotes the probability of dropping a word from the transcript label. It can be seen that higher value of $p\_drop$ corresponds to higher number of missing words in the transcript and vice-versa. Figures 7a-b show the histogram of percentage of the labeled words present in each training sample in LibriSpeech for different values of $p\_drop$ which closely follows a normal distribution. Also, from the definition of $p\_drop$, it naturally follows that $p\_drop = 0$ corresponds to supervised training setup while $p\_drop = 1$ corresponds to unsupervised training.

To make sure our method is robust to different types of partial labels, we also test our method on two other ways of generating the data :
1. Randomly split the training set into $p$ parts and use a different $p\_drop$ for each split (Figure 7c)
2. Randomly split the set of all words in training vocab into $p$ parts, use a different $p\_drop$ for each split (Figure 7d)

---

[3]`https://github.com/flashlight/flashlight`

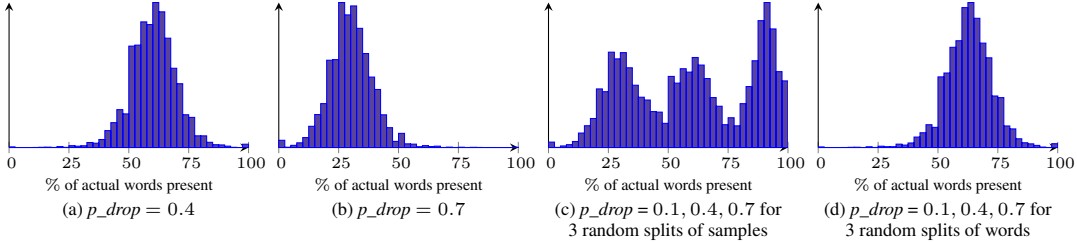

| (a) $p\_drop = 0.4$ | (b) $p\_drop = 0.7$ | (c) $p\_drop = 0.1, 0.4, 0.7$ for 3 random splits of samples | (d) $p\_drop = 0.1, 0.4, 0.7$ for 3 random splits of words |

Figure 7: Histograms of percentage of transcript words retained in each training sample in LibriSpeech for different ways of generating weakly supervised data using $p\_drop$.

For all the new partially labeled datasets created, we prune samples which have empty transcriptions before using them for training.

### 4.1.2 Pseudo-labeling

To further achieve better word error rate (WER), we generate pseudo labels (PLs) on the training set using the word-based model trained with STC and use these PLs to train a letter-based model, via a regular CTC approach. The PLs are generated using rescoring as described in Section B.1.2 on the training set using the hyperparameters optimized on validation set.

## 4.2 Handwriting Recognition (HWR)

**True Label :** once-and-for-all|cash|offer|of|357million|.|President

*pDrop = 0.1 :* oc-andforall|cah|offer|o|357million|.|Preident

*pDrop = 0.3 :* once-nd-fralcasr|of|357milon|Presidnt

*pDrop = 0.7 :* nc-llaf|7mlioPreit

Figure 8: A training example from IAM along with weakly supervised labels generated for different $p\_drop$ values.

We test our approach on IAM Handwriting database [26], which is a widely used benchmark for handwriting recognition. The dataset contains 78 different characters and a white-space symbol, |. We use Aachen data splits[4] to divide the dataset into three subsets: 6,482 lines for training, 976 lines for validation and 2,915 lines for testing.

To create weakly supervised labels, we use the same methodology using $p\_drop$ as described in 4.1.1. However, we drop the labels at character-level instead of word-level. This is because it is efficient to express the STC label graph with "$\langle * \rangle$" tokens when we use the same type of targets which are being dropped. In ASR, the targets are dropped at word level, hence we choose words and in HWR we use character targets. Figure 8 shows a training example from IAM along with weakly supervised labels generated for different $p\_drop$ values.

Additional implementation and training details (models, tokens, optimization, other hyperparameters and settings) for the ASR and HWR experiments are in the Appendix B.

---

[4]https://www.openslr.org/56

Table 1: WER comparison on LibriSpeech test sets for various partial label generation settings. For each setting, we report the results for training with CTC (control experiment), STC on the partially labeled data and training with CTC using PLs generated from the trained STC model. Whenever possible, for each of these experiments, we report WER with a greedy decoding and no LM (top row), with 5-gram LM beam-search decoding (middle row) and with additional second-pass rescoring by Transformer LM (below row). We also include state-of-the-art results on LibriSpeech using wordpieces, letters and words as output tokens in the top section of the table.

| Weak label gen. Strategy | Method | Criterion | Model Stride/ Parameters | Output Tokens | LM | Test WER clean/other |
|---|---|---|---|---|---|---|
| | Conformer [14] | RNN-T | 4/119M | 1K Word Pieces | - 
 LSTM | 2.1/4.3 
 1.9/3.9 |
| | Transformer [36] | CTC | 3/270M | Letters | - 
 Word 5-gram 
 Tr. Rescoring | 2.7/6.1 
 2.4/5.3 
 2.1/4.5 |
| | Transformer [8] | seq2seq | 8/~300M | Words | 
 Word 4-gram | 2.9/6.7 
 3.0/6.3 |
| p_drop = 0.1 | Transformer | CTC | 3/270M | Letters | - 
 Word 5-gram 
 Tr. Rescoring | 6.2/11.0 
 4.9/8.8 
 4.5/8.1 |
| | Transformer | STC | 8/70M | Words | - 
 Word 5-gram 
 Tr. Rescoring | 4.1/9.0 
 4.1/8.7 
 3.3/6.9 |
| | + Pseudo Labeling | CTC | 3/270M | Letters | - 
 Word 5-gram 
 Tr. Rescoring | 3.1/6.3 
 2.9/5.6 
 2.7/5.2 |
| p_drop = 0.4 | Transformer | CTC | 3/270M | Letters | - | 48.2/56.9 |
| | Transformer | STC | 8/70M | Words | - 
 Word 5-gram 
 Tr. Rescoring | 4.5/10.2 
 4.6/9.9 
 3.6/7.7 |
| | + Pseudo Labeling | CTC | 3/270M | Letters | - 
 Word 5-gram 
 Tr. Rescoring | 3.3/6.6 
 3.1/5.9 
 2.9/5.4 |
| p_drop = 0.7 | Transformer | CTC | 3/270M | Letters | - | 100/100 |
| | Transformer | STC | 8/70M | Words | - 
 Word 5-gram 
 Tr. Rescoring | 6.9/15.1 
 7.0/14.8 
 5.1/11.1 |
| | + Pseudo Labeling | CTC | 3/270M | Letters | - 
 Word 5-gram 
 Tr. Rescoring | 3.9/7.9 
 3.5/7.0 
 3.1/6.2 |
| Split all samples into 3 parts randomly; Assign p_drop= 0.1,0.4,0.7 for the splits | Transformer | CTC | 3/270M | Letters | - | 58.6/63.9 |
| | Transformer | STC | 8/70M | Words | - 
 Word 5-gram 
 Tr. Rescoring | 4.9/11.1 
 5.1/11.0 
 3.7/8.2 |
| | + Pseudo Labeling | CTC | 3/270M | Letters | - 
 Word 5-gram 
 Tr. Rescoring | 3.3/6.7 
 3.0/5.9 
 2.8/5.4 |
| Split all words into 3 parts randomly; Assign p_drop= 0.1,0.4,0.7 for the splits | Transformer | CTC | 3/270M | Letters | - | 45.6/49.2 |
| | Transformer | STC | 8/70M | Words | - 
 Word 5-gram 
 Tr. Rescoring | 5.8/13.0 
 6.6/12.2 
 4.1/8.7 |
| | + Pseudo Labeling | CTC | 3/270M | Letters | - 
 Word 5-gram 
 Tr. Rescoring | 3.4/6.9 
 3.1/5.9 
 2.9/5.6 |

## 5 Results

### 5.1 Automatic Speech Recognition

In Table 1, we compare the WER performance of STC models for a fixed value of $p\_drop = 0.1, 0.4, 0.7$ and also when $p\_drop$ is dependent on the sample or the word. We also compare these results with the models trained directly with CTC on the partial labels. We can see that STC performs better than CTC in all the settings. We observe that performing beam search decoding

| P_DROP | METHOD | DEV CER | TEST CER |
|---|---|---|---|
| | TRANSFORMER [21] | | 4.7 |
| | CNN+CTC [43] | | 4.9 |
| | LSTM W/ATTN [28] | 3.2 | 5.5 |
| | CNN + CTC (OUR BASELINE) | 3.7 | 5.4 |
| 0.1 | CNN + CTC | 5.4 | 7.7 |
| | CNN + STC | 5.0 | 7.2 |
| 0.3 | CNN + CTC | 8.7 | 11.6 |
| | CNN + STC | 5.6 | 8.1 |
| 0.5 | CNN + CTC | 48.2 | 53.6 |
| | CNN + STC | 10.0 | 13.5 |
| 0.7 | CNN + CTC | 77.3 | 78.5 |
| | CNN + STC | 22.7 | 26.7 |

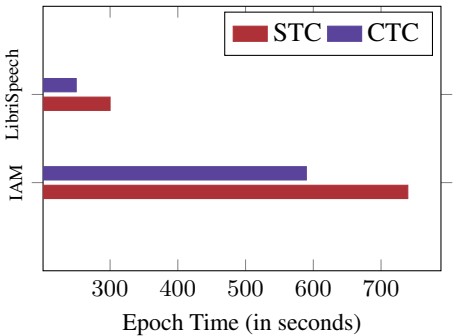

Table 2: CER performance comparison of STC and CTC with greedy decoding on IAM dataset for different $p\_drop$ values. A comparison with selected works using the "same" training set is shown at the top.

Figure 9: Epoch time performance of STC, CTC on LibriSpeech and IAM datasets for the models we have used.

and an additional second-pass rescoring with a Transformer LM can reduce the WER of the models significantly.

We also see that pseudo labeling the training set using STC model trained on partial labels and then training a letter-based CTC model can further improve WER performance. We can get competitive results compared with the fully supervised results.

## 5.2 Handwriting Recognition

In Table 2, we compare the CER performance of CTC and STC for various values of $p\_drop$ with greedy decoding. We can see that using STC clearly gives better performance over CTC trained models for $p\_drop = 0.1, 0.3, 0.5, 0.7$. The performance gap between the supervised baseline and models trained with partial labels using STC is larger compared to ASR experiments. This is because we did not use LM decoding or pseudo-labeling steps which should help in reducing CER.

Hyperparameters used for token insertion penalty, $\lambda$ to train STC models for ASR, HWR experiments are in Appendix D.

## 5.3 Runtime Performance of STC

In Figure 9, we compare the epoch time of STC, CTC models on LibriSpeech and IAM datasets. For a fair comparison, we use the same model that is used to report STC results for each dataset. The experiments on LibriSpeech, IAM are run on 32, 8 Nvidia 32GB V100 GPUs and uses C++, Python APIs of GTN respectively.

We were able to run an epoch on LibriSpeech in about 300 seconds for the STC model and in about 250 seconds for the CTC model. Optimizing the data transfer between CPU and GPU by only moving the tokens which are present in the current training sample (Section 3.3.3) is a crucial step in achieving good performance. Without this optimization, it would take about 5400 seconds per epoch on LibriSpeech as the alphabet size, $|\mathcal{A}|$ of 50000 is very high.

For IAM experiments, we did not perform the data transfer optimization for STC as $|\mathcal{A}|$ is only 79. It can be see that the epoch time of STC is about 25% more compared to CTC.

## 6 Conclusion

In a variety of sequence labeling tasks (ASR, HWR), we show that STC enables training models with partially labeled data and can give strong performance. Weakly supervised data can be collected in semi-automatic ways and can alleviate labeling of full training data, thus reducing production costs.

We also show that using WFSTs with a differentiable framework is flexible and powerful tool for researchers to solve a completely new set of problems, like the example we have shown in this paper. STC requires additional hyperparameter tuning and these vary by $p\_drop$. The label graph can be easily modified to include the cases where the missing label's position or the number of missing tokens is known in advance. Also, STC can be specialized to allow a particular range of missing tokens (e.g. between 2 and 4 characters). A direct potential extension is to study the application of STC to noisy labels.

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
