# OpenReview forum: "Star Temporal Classification: Sequence Modeling with Partially Labeled Data"
_NeurIPS.cc/2022/Conference — NeurIPS 2022 Accept_

### Official Review · Reviewer_twp9 · 2022-07-10

**Rating:** 6
**Confidence:** 4
**Soundness:** 2 fair
**Presentation:** 3 good
**Contribution:** 2 fair

**Summary:**

The paper proposes Star Temporal Classification (STC) to solve the problem of missing labels for weakly supervised sequence labeling tasks (such as ASR, HWR). STC uses a special star token to allow alignments which include all possible tokens whenever a token could be missing. Technically, STC uses differentiable WFSTs to implement the loss function. Experiments are conducted for speech recognition over Librispeech and handwriting recognition over IAM, which demonstrate the effectiveness of STC.


**Questions:**

see above

**Limitations:**

see above

**Strengths And Weaknesses:**

-Strengths

The paper is generally well-written. The problem of missing labels is clearly formulated and solved in the framework of differentiable WFSTs. The techniques of introducing the star token and token insertion penalty are new. Moreover, the implementation of STC, especially to integrate CPU-based WFST algorithm and GPU-based neural network gradient calculation, is non-trivial, which requires engineering efforts. The experiments seem to be good.

-Weaknesses

Concerns about the experiments:

1) The proposed method is designed to work for weakly supervised setup. But in practice, it is not known whether the labels are missing and the proportion of missing is also unknown. So the performance of applying STC in the fully-labeled setting is an important indicator. If such performance becomes worse due to the use of STC, then STC may not be practically useful.

2) In real-world tasks, the labels are missing for many reasons. An important scenario is that the labels are not really missing, but are discarded after some filtering such as by confidence measures. This scenario is described in Line 21-23 in the Introduction, but not fully evaluated in the paper.

The proposed method is only experimented in the simple scenario, where some positions in the label sequence are really missing (not labeled). This is a simple scenario, which may not be so interesting in real-world tasks. The baseline of running CTC over the incomplete labels is too weak.

I suggest that the authors can take experiments and show improvement in more realistic scenarios (e.g., semi-supervised learning or transfer learning), as described in the Introduction.

---

> ### Author Response · Authors · 2022-08-02
> **Responses to Reviewer twp9**
>
> Thank you so much for your positive feedback and comments. We would like to answer your questions as follows:
>
> **The proposed method is designed to work for weakly supervised setup. But in practice, it is not known whether the labels are missing and the proportion of missing is also unknown. So the performance of applying STC in the fully-labeled setting is an important indicator. If such performance becomes worse due to the use of STC, then STC may not be practically useful.**
>
> Training STC with pDrop=0.0 corresponds to training a CTC model if we keep token insertion penalty = -inf.  We have added more details in discussion of the top-level comment under “STC with pDrop=0.0”.
>
> **The proposed method is only experimented in the simple scenario, where some positions in the label sequence are really missing (not labeled). This is a simple scenario, which may not be so interesting in real-world tasks. The baseline of running CTC over the incomplete labels is too weak.**
>
> **In real-world tasks, the labels are missing for many reasons. An important scenario is that the labels are not really missing, but are discarded after some filtering such as by confidence measures. This scenario is described in Line 21-23 in the Introduction, but not fully evaluated in the paper.**
>
> **I suggest that the authors can take experiments and show improvement in more realistic scenarios (e.g., semi-supervised learning or transfer learning)**
>
>
> While we agree that we use simulated data, we do not believe we are using simple scenarios for running the experiments. We have measured the performance for various values of pDrop (upto 70% of labels missing). For ASR task, we have also tested scenarios where pDrop varies per sample / per word.
>
>
> We have provided strong baselines (comparable to SOTA) on both LibriSpeech/IAM and using simulated label removal. The same training recipes have been used for running all the partial label experiments.
>
> We have also added some discussion in the top-level comment under “Experiments on Simulated Data/Concerns about baselines”.

---

> > ### Author Response · Authors · 2022-08-09
> > **Followup**
> >
> > Dear reviewer,
> >
> > Thank you again for all the valuable comments and suggestions. We answered the questions and provided additional materials as above. We hope these address your concerns and please take them into consideration for the final scores. We are happy to answer additional questions.

---

### Official Review · Reviewer_eojv · 2022-07-21

**Rating:** 6
**Confidence:** 4
**Soundness:** 3 good
**Presentation:** 3 good
**Contribution:** 3 good

**Summary:**

This paper presents Star Temporal Classification (STC) an approach to train sequential models where there is an expectation that some labels are missing.  The key machinery to accomplish this is composition with a WFST constructed, and appropriately weighted to train with sequences of the (regular expression) form * A * B * C * rather than A B C.

**Questions:**

* The introduction and abstract experimental performance is described imprecisely. "recover most of the performance" "low word error rates".  It might be more impactful to be more exact in these descriptions -- how close to fully observed labeling exactly? what does "low" mean in these contexts.

* STC is not evaluated with pDrop=0.0. This data point would provide confidence that this approach can be used without losing performance if labels are present (or if they are not).  Even the randomly split samples always assume at least 10% of labels are dropped.

* What is the impact of modifying the collapse function to only remove blanks instead of also removing duplicates?

* How is the token insertion penalty tuned? How big an impact does correctly setting this hyperparameter have?

* What is the rationale for word based targets followed by letter based refinement?  The 1k word pieces seems to be a better target for librispeech asr, but this isn't used in any CTC or STC experiments. why not?

* re baselines: Why not use a Conformer model in these experiments?  (the rationale for using CTC rather than RNNT is clear though.)

* The best performing approach for STC requires a two-pass training routine using Word targets to recover missing labels and then Letter targets for final training. The word targets are described as an efficiency bottleneck in section 5.3 as well.  Why can't STC work on letter targets directly?  Is this because the sequence length is too long with letter targets?  Is there any benefit to this style training for the CTC baseline?  While the performance improvements are clear, this seems unrelated to the central STC idea.

* A couple of possible typos in table 1.  The second to last line reports errors of ".1/5.9".  The third row in the LM column includes "2.5/5.9"

**Limitations:**

* Without an evaluation at pDrop=0.0 it is hard to have confidence that this is a reliable drop-in replacement for CTC training.  If it's not this limits the impact of this work to only those instances where a user has a reasonable understanding that the available training data has been partially labeled.

* STC requires additional hyperparameter tuning and these vary by pDrop.

**Strengths And Weaknesses:**

Strengths
* STC is a clearly written extension to CTC, leveraging the ability to construct a decoding topology with a WFST.
* The basic approach taken opens itself up to modifications of the WFST beyond the STC formulation.
* Demonstrates that STC can substantially improve performance on ASR and handwriting recognition, even with relatively high rates of label deletion.

Weakness
* The practical motivation is somewhat strained. While there are plenty of instances where labels may be missing (occlusion etc.), however, in most (many?) cases including those offered in the introduction the "missing" label's position is known.  For example for semi-supervised learning to mask low-confidence tokens, while the identity of the tokens may not be trusted, their position usually can.  Similarly in the handwriting recognition example, while there may be ambiguity in which characters are present, the number of characters is less confusable.

---

> ### Author Response · Authors · 2022-08-02
> **Responses to Reviewer eojv**
>
> Thank you so much for your positive feedback and we would like to answer your questions as follows:
>
>
> **The practical motivation is somewhat strained. While there are plenty of instances where labels may be missing (occlusion etc.), however, in most (many?) cases including those offered in the introduction the "missing" label's position is known.**
>
> The label graph can be easily modified to include the cases where the missing label's position or the number of missing tokens is known in advance. In fact the idea for STC came from trying to solve a similar problem where we know the number of characters missing, and in these cases we noticed that there is no need to have a token insertion penalty. Also, STC can be specialized to allow a particular range of missing tokens (e.g. between 2 and 4 characters).
>
> In this work, our aim is to develop STC as a general framework for solving missing label problems. However, customizing STC to the problem specific requirements is certainly possible.
>
> **The introduction and abstract experimental performance is described imprecisely. "recover most of the performance" "low word error rates"**
>
> Thank you, we have added details to this section in an attempt to make it clearer.
>
> **STC is not evaluated with pDrop=0.0. This data point would provide confidence that this approach can be used without losing performance if labels are present (or if they are not). Even the randomly split samples always assume at least 10% of labels are dropped.**
> **Without an evaluation at pDrop=0.0 it is hard to have confidence that this is a reliable drop-in replacement for CTC training. If it's not this limits the impact of this work to only those instances where a user has a reasonable understanding that the available training data has been partially labeled.**
>
> Please see the discussion in the top-level comment under “STC with pDrop=0.0”.
>
> **What is the impact of modifying the collapse function to only remove blanks instead of also removing duplicates?**
>
> In the STC label graph, we use star tokens ( <s> and <s>\t) to reduce the number of arcs present in the WFST graph so that WFST operations can be done efficiently (as discussed in Figure 3 from paper).
>
> In order to use the star tokens, we had to remove self-loops on non-blank tokens. Otherwise, the same alignment can occur on multiple paths in the STC and label composed graph. Disallowing self-loops on non-blank tokens during training implies a collapse function which removes only blank tokens.
>
> Here is an example to understand the  multiple path issue. Consider that we have a self-loop arc with label “b”  at node 3 in Figure 4 of the paper. Now, when the input is “b” at node 3, the WFST can either accept the self-loop arc or accept the arc from node 3-> node 5 (note that <s> include “b” as well). Thus, we can end-up counting the same sequence multiple times while composing with the graph.
>
> **Why can't STC work on letter targets directly? Is this because the sequence length is too long with letter targets? Is there any benefit to this style training for the CTC baseline? While the performance improvements are clear, this seems unrelated to the central STC idea.
> What is the rationale for word based targets followed by letter based refinement?**
>
> It is efficient to express the STC label graph with star tokens when we use the same type of targets which are being dropped. In ASR, the targets are dropped at word level, hence we choose words and in HWR we use letter targets.
> And the reason for doing this is because creating the label graph by making use of star tokens to reduce graph size becomes difficult otherwise. The main challenge here being not to make the label WFST graph accept the same alignment path multiple times.
>
> **How is the token insertion penalty tuned? How big an impact does correctly setting this hyperparameter have?**
>
> We have also added a new section in the appendix where we show the sensitivity of the model performance with respect to this hyperparameter.
>
> **The 1k word pieces seems to be a better target for librispeech asr, but this isn't used in any CTC or STC experiments. why not?**
>
> **re baselines: Why not use a Conformer model in these experiments? (the rationale for using CTC rather than RNNT is clear though.)**
>
> For training with CTC on LibriSpeech, the best published result so far uses Transformer models with letters and we have used the same recipe for our experiments. While we may be able to get similar/better performance with Conformers, it may require tuning the model architecture and it is beyond the scope of this paper. We have provided strong baselines (comparable to SOTA) on LibriSpeech.
>
>
> **A couple of possible typos in table 1. The second to last line reports errors of ".1/5.9". The third row in the LM column includes "2.5/5.9"**
>
> Thanks for catching the typos. We have addressed these issues in the newer version.

---

> > ### Author Response · Authors · 2022-08-09
> > **Followup**
> >
> > Dear reviewer,
> >
> > Thank you again for all the valuable comments and suggestions. We answered the questions and provided additional materials as above. We hope these address your concerns and please take them into consideration for the final scores. We are happy to answer additional questions.

---

### Official Review · Reviewer_dZmF · 2022-07-23

**Rating:** 4
**Confidence:** 4
**Soundness:** 2 fair
**Presentation:** 2 fair
**Contribution:** 2 fair

**Summary:**

This paper proposes star temporal classification (STC), an extension of CTC particularly designed when only partial labeled supervision is available.

**Questions:**

I would like to hear authors opinion about my points in weakness.

**Limitations:**

I did not see any discussion on limitations.

**Strengths And Weaknesses:**

Strength:
- Modeling when partial label is available is an interesting problem and can have several applications including the ones explained in introduction. A very relevant example can be recovering complete text for ancient literature when only partial text is recovered (references 2 and 3 of paper).

Weakness:
- One major issue with this paper is clarity of text and definitions. Examples:
1. Use of <s> as star token: this symbol is widely used as start-of-sentence in the literature
2. Equation 3, $P(<s>|x) = \sum_{y\in\mathcal{A}} P(y|x)$, I think there should be some notion of time in this equation, otherwise I am not sure if rest of model makes sense. Second, if this is time dependent posterior, is it simply 1.0 - P(<b> | x, t)?
3. text issues:page 5, "is is useful think ..."
4. Many issues in Table1:
4.a row starting with "TRANSFORMER[36]": there is some number under LM (2.5/5.9), what does this mean?
4.b one to the last row, there is 0.1/5.9, does this model perform 0.1 on clean or is it a typo?
5. The definition of the randomly split the words in bottom of page 6 is not clear to me.
6. The notation pDrop, how about p_drop ?

- Another major concern I have is the significance of modeling presented in this paper.
I think the change is not very far from original CTC, it only changes state transition of CTC, this can be done for any other model, like RNNT. While important, I am not sure if it is significant.

- Finally, there is a major concern about baseline comparisons:
1. All the experiments presented are on simulated dataset created by authors, no previous numbers are reported on these partially labeled dataset. The main comparison is with fully labeled data. I don't think this sufficiently evaluate STC. Why not reporting some number on some other datasets, like the ones presented in reference 2 or 3.
2. STC is not compared against other methods for recovering partially labeled data. For example one baseline can be creating pseudo label from CTC + LM for missing labeled and train with partial+pseudo labels. Maybe this will do as good as STC?

---

> ### Author Response · Authors · 2022-08-02
> **Responses to Reviewer dZmF**
>
> Thank you for your valuable feedback and comments. We would like to answer your questions here.
>
> **Use of <s> as star token: this symbol is widely used as start-of-sentence in the literature**
>
> **The notation pDrop, how about p_drop ?**
>
> **I think there should be some notion of time in this equation, otherwise I am not sure if rest of model makes sense. Second, if this is time dependent posterior, is it simply 1.0 - P(<b> | x, t)**
>
> **text issues:page 5, "is is useful think ..."**
>
> **Many issues in Table1: 4.a row starting with "TRANSFORMER[36]": there is some number under LM (2.5/5.9), what does this mean? 4.b one to the last row, there is 0.1/5.9, does this model perform 0.1 on clean or is it a typo?**
>
> Thank you for your corrections and suggestions regarding equations and notation. We have made modifications in the manuscript accordingly.
> Also, P(<s> | x, t) = 1.0 - P(<b> | x, t) holds true.
>
>
> **The definition of the randomly split the words in bottom of page 6 is not clear to me.**
>
> We consider the set of all the words in training vocab and split them into 3 groups randomly. Then, we assign pDrop = 0.1 for the first group,  pDrop = 0.4 for the second group and pDrop = 0.7 for the third group. Now given a fully labeled sentence, we drop each word with a probability based on pDrop assigned to its group.
>
>
> **Another major concern I have is the significance of modeling presented in this paper. I think the change is not very far from original CTC, it only changes state transition of CTC, this can be done for any other model, like RNNT. While important, I am not sure if it is significant.**
>
> The STC algorithm improves upon CTC and is different from it in several ways. First, we note that the STC algorithm shows substantial improvements in settings for which CTC completely fails. Second, we show how to express STC using WFSTs which makes it a general and extensible framework for partial labels. Finally, getting STC to work well in practice required non-trivial changes to both the graphs involved and optimizations to make the algorithm tractable.
>
> We also point out a few papers with extensions to CTC for weakly supervised modeling that have been published in the past:
>
> “Connectionist Temporal Modeling for Weakly Supervised Action Labeling” published in ECCV 2016
>
> “W-CTC: a Connectionist Temporal Classification Loss with Wild Cards” published in ICLR 2022.
>
>
>
> **All the experiments presented are on simulated dataset created by authors, no previous numbers are reported on these partially labeled dataset. The main comparison is with fully labeled data. I don't think this sufficiently evaluate STC. Why not reporting some number on some other datasets, like the ones presented in reference 2 or 3.**
>
> Please see the discussion in the top-level comment under “Experiments on Simulated Data/Concerns about baselines”.
>
>
> **STC is not compared against other methods for recovering partially labeled data. For example one baseline can be creating pseudo label from CTC + LM for missing labeled and train with partial+pseudo labels. Maybe this will do as good as STC?**
>
> We agree with the reviewer that this would be a good baseline to compare with. But regarding the question on “CTC + LM will do as good as STC”, we think using STC+LM has a clear advantage over CTC+LM. We can see that STC+LM can generate better pseudo labels than CTC+LM in all scenarios. It might also be interesting to see if iterative pseudo labeling for CTC+LM can reduce the gap, but that is beyond the scope of this paper. Also, with higher pDrop values, for example pDrop=0.7, CTC models achieve a WER of 100% and hence cannot generate useful pseudo labels.
>
>
> **Limitations not specified**
>
> From our perspective, one limitation of STC is the need to tune the token insertion penalty. We have added this to the paper.
>
> We have also added a new section in the appendix where we show the sensitivity of the model performance with respect to this hyperparameter.

---

> > ### Author Response · Authors · 2022-08-09
> > **Follow up**
> >
> > Dear reviewer,
> >
> > Thank you again for all the valuable comments and suggestions. We answered the questions and provided additional materials as above. We hope these address your concerns and please take them into consideration for the final scores. We are happy to answer additional questions.

---

### Author Response · Authors · 2022-08-02
**Top level response to reviewers**

Dear reviewers,

Thank you for your constructive feedback. We also very much appreciate the positive feedback including comments such as this work could “*open itself up to modifications of the WFST beyond the STC formulation*” and the “implementation of STC, especially to integrate CPU-based WFST algorithm and GPU-based neural network gradient calculation, is non-trivial, which requires engineering efforts”.

We have updated the paper based on some of the suggestions that were given.

We would like to address some comments that were common amongst the reviewers at the start.

**STC with pDrop=0.0**

The label graph of STC is built upon CTC and allows paths for missing tokens in between labels. The weights of these additional paths is determined by the value of “token insertion penalty”. Since it is a hyperparameter, we can always set it to -inf and the resulting graph becomes similar to CTC (with self-loops on non-blank tokens removed). We have provided CTC baselines for all experiments.

**Experiments on Simulated Data/Concerns about baselines**

We agree that a real-world benchmark would substantiate the performance of STC. However, we do not know of the existence of an established academic speech recognition or handwriting recognition benchmark which exhibits partial labeling. In the absence of such a benchmark, we have provided strong baselines (comparable to SOTA) on both LibriSpeech/IAM and using simulated label removal under various experimental setups. Currently we leave the construction of a more realistic academic benchmark for partial labeling as an important direction for future work.

---

### Meta-Review · Area_Chair_aQSu · 2022-09-09

**Recommendation:** Accept
**Confidence:** Less certain

**Metareview:**

This paper is right on the border. I'm going to mark this as accept as the only reviewer marking reject is due to limited evaluation, but i believe the evaluation is ok (as do the other two reviewers). The idea is interesting and novel, and the paper is well written.

**Award:**

No

---

### Decision · Program_Chairs · 2022-09-14

Accept